# A Systematic Review of NeurIPS Dataset Management Practices

**Yiwei Wu**
University of Texas at Austin
yw25689@utexas.edu

**Leah Ajmani**
University of Minnesota
ajman004@umn.edu

**Shayne Longpre**
MIT
slongpre@media.mit.edu

**Hanlin Li**
University of Texas at Austin
lihanlin@utexas.edu

## Abstract

As new machine learning methods demand larger training datasets, researchers and developers face significant challenges in dataset management. Although ethics reviews, documentation, and checklists have been established, it remains uncertain whether consistent dataset management practices exist across the community. This lack of a comprehensive overview hinders our ability to diagnose and address fundamental tensions and ethical issues related to managing large datasets. We present a systematic review of datasets published at the NeurIPS Datasets and Benchmarks track, focusing on four key aspects: provenance, distribution, ethical disclosure, and licensing. Our findings reveal that dataset provenance is often unclear due to ambiguous filtering and curation processes. Additionally, a variety of sites are used for dataset hosting, but only a few offer structured metadata and version control. These inconsistencies underscore the urgent need for standardized data infrastructures for the publication and management of datasets.

## 1 Introduction

Datasets serve as a fundamental bedrock for machine learning models. While the construction of larger datasets has propelled advancements in the field, it has also revealed negative implications of these practices [33]. Issues include a lack of consent and compensation [2], documentation debt [4, 17], legal risks [31], and malign content [6, 5, 13, 12].

To incentivize responsible management of datasets, the research community proposed various procedures, e.g. ethics review[19], dataset licensing [11], and standardized dataset documentation [17]. While these toolkits have promoted individual reflections to be more conscious of ethical and legal issues in sharing datasets, limited work examined the practices of data management at scale [42]. To what extent do data management practices align or vary among researchers?

The lack of a comprehensive overview of data management practices hampers our ability to systematically diagnose and address key tensions in managing large datasets. To tackle this issue, we reviewed

The 38th Conference on Neural Information Processing Systems (NeurIPS 2024) Track on Datasets and Benchmarks. Do not distribute.

238 dataset papers published in the NeurIPS Datasets and Benchmarks track, identifying significant gaps and inconsistencies in how researchers and developers share and manage their datasets. Given the exponential growth of the track and the increasing need for effective dataset stewardship in machine learning [8], we focused on four critical aspects of data management where publishers and stakeholders can have an immediate impact: provenance, distribution, ethical disclosure, and licensing.

We found major inconsistencies in all four key aspects, as well as some dominant trends. 57% papers were built off of existing datasets or external data sources, but the level of detail on access, data filtering, and curation varies greatly, making it difficult to trace provenance at times. Datasets were distributed on a variety of sites, such as personal or group websites, GitHub, and Zenodo, each offering different levels of support for metadata and version control. Some authors discussed their ethical considerations extensively, while most did not identify any potential implications. With respect to licensing, Creative Commons licenses were the predominant choices, but there were exceptions in which the authors did not specify a license for their public datasets. Moreover, important dataset terms and license information appear in different locations of a dataset project, e.g. in the body of a paper, in supplemental materials, or on the dataset's hosting site, making it difficult for potential dataset users to gain an accurate understanding of permissible use cases and limitations.

Our findings on data authors' differing practices and choices on the four aspects highlight the urgent need for standardized data infrastructures to support the remixing and sharing of datasets with clear metadata, ethical disclosures, and complete licensing information. Dataset authors need not only structured documentation template but also standardized data infrastructures for hosting. As datasets continue to lay the foundation for large models, publishers, research institutions, and funding agencies should join forces and prioritize developing and promoting such data infrastructures so that datasets can be responsibly and ethically shared, managed, and used by researchers and practitioners. We also discuss our findings in the context of the FAIR principles – widely accepted rules of scientific data management (findable, accessible, interoperable, and reusable) [40] – and scholarly discourse on licensing and provide recommendations for dataset authors on sharing datasets.

## 2 Methods

We focused on dataset papers published at the NeurIPS Datasets and Benchmarks track. NeurIPS's prominence in the ML field and the track's emphasis on dataset contributions make these dataset papers suitable cases for our investigation into dataset management practices. We excluded papers that solely focused on contributing or comparing benchmarks without introducing any new datasets. We reviewed all dataset papers published in 2021 (73 papers) and 2022 (85 papers). Notably, 2023 saw substantial growth in number of dataset papers and we randomly sampled 80 out of a total of the 193 published. Because our intended contribution is to capture differences dataset management practice rather than quantitatively mapping out the distribution of specific practices, we began with sampling 20 papers of the 2023 batch. We reached qualitative saturation and temporal balance at 80 papers from 2023.

Our review focused on four key aspects of data management practices: 1) **Provenance**: Is it possible to trace provenance? 2) **Distribution**: How was the dataset distributed? 3) **Ethical Disclosures**: What ethical concerns did the authors disclose? 4) **Licensing**: Under what licenses were the datasets released?

The first and last authors went through 30 randomly selected papers to establish a schema to categorize papers on each aspect. Then the two authors annotated the rest of the dataset papers independently and periodically cross-checked each other's labels. The authors also met regularly to iteratively revise categories and labels and annotated edge cases after reaching a consensus. The full list of dataset papers along with our annotations can be found at https://doi.org/10.18738/T8/HLTRQP, hosted by the Texas Data Repository.

# 3 Results

The 238 papers contribute datasets of a variety of modalities, including images, texts, and action logs, spanning different domains, e.g. health, banking, biology, and geography. The topical and methodological varieties of the reviewed datasets highlight the Datasets and Benchmarks track's broad appeal to ML researchers, making the venue a suitable subject for our review.

Our goal is not to critique any existing practices but rather to identify differences and inconsistencies to support the improvement of existing guidelines, processes, and infrastructures, so researchers can build, remix, and expand datasets ethically and responsibly, a key goal for the track [10].

## 3.1 Is it possible to trace provenance?

We examined the possibility of tracing provenance and found a stark contrast among different data collection methods. Here, we adopted Longpre et al's[28] definition of provenance as considerations relevant to a dataset's original source and creation. These are crucial considerations for the development of accurate, responsible AI models. We find a wide range of data collection techniques, each with its own provenance considerations. Below we describe the four distinct approaches authors take to construct their datasets–post hoc, ad hoc, synthesis, and annotation–and how provenance may or may not be an issue in each approach. Note that these four categories are not mutually exclusive.

### 3.1.1 Post hoc

Post hoc refers to when authors collect data from other entities after it has been generated and recorded, typically through direct downloads, scraping, API, or data sharing agreements. 136/238 (57%) papers fall under the post hoc category.

Tracing provenance is a difficult task for the post hoc category, despite the vast majority of these papers (120) relied on publicly available data. When describing their specific data collection approach, authors used a variety of descriptions, e.g. exporting, downloading, requesting, and collecting. The ambiguity of the language makes it difficult for those outside of the research team to know whether the raw data is curated through direct downloads, API, or scraping and whether the dataset authors had to clean or structure the raw data (a procedure that is typically required for scraped data).

Furthermore, the original data curation and preprocessing procedures are often unclear; we noticed missing details about manual curation criteria, limited documentation about processing, and in some instances authors' wish to keep sensitive information in the original source private. The lack of information about curation and preprocessing could be a provenance issue for potential data users who are interested in examining the source of original data points.

16 papers included data collected post hoc from private servers or mentioned collaborations with external organizations, including imaging companies, biotech companies, hospitals, and government agencies. Given the private nature of the original data, provenance is inherently challenging. However, some authors provided background information about the collaborating entity, their processing and filtering procedures, and in some cases, authors even shared the protocol or information about the instrument used by collaborators, e.g. [29].

### 3.1.2 Ad hoc

Ad hoc refers to when the authors developed the instrument or apparatus to collect data from subjects directly. 72 papers in our review fall under this category. Because it is the authors who created the data or guided the data creation, tracing provenance for such type of dataset is naturally straightforward. For papers under this category, authors generally provided extensive details discussing their specific mechanisms, e.g. web interface, sensing technologies, camera setup, etc. 31 papers collected data from individuals or environments without researchers providing instructions and interventions, i.e. "in the wild". 41 papers collected data "in the lab" where specific instructions/interventions were given to participants.

Table 1: Hosting sites used by over 5 dataset papers.

| Hosting sites | Number of dataset papers |
|---|---|
| Personal or lab websites (including those hosted on github.io) | 44 |
| Zenodo | 34 |
| GitHub | 33 |
| Google Drive | 32 |
| Hugging Face | 20 |
| Institutional data storage service, e.g. digital library | 13 |
| Kaggle | 8 |
| Baidu Drive | 7 |
| AWS | 6 |
| Physionet | 5 |

### 3.1.3 Synthesis

39 papers in our review included synthesized datasets. We define synthesis as authors substantially transforming original datasets or creating new datasets with instruments. Therefore, synthesized datasets are different from simply combining or aggregating datasets. While synthetic datasets were artificially created by authors, provenance could still be an issue when synthetic datasets are built upon existing datasets without suffcent documentation about the transformation. For example, two papers in our review lacked description on data sampling and potential dataset users will likely find it difficult to verify the data curation procedure.

### 3.1.4 Annotation

The annotation category includes dataset papers that involved human annotators in constructing datasets. 91 papers fall under this category. Provenance of annotations is typically clear at a high level because authors generally disclose who are the annotators or how they are recruited. In our review, 41 papers involved crowd workers, 36 involved experts, and 19 papers involved authors themselves as annotators. Note that one paper may employ different types of annotators.

However, 13 out of 41 papers that used crowdworkers did not specify the platform or site used for recruiting workers. This may become a provenance issue for potential dataset users who wish to verify the working conditions of crowd workers.

### 3.2 How were datasets distributed?

Table 1 provides an overview of the platforms and services authors used to distribute their datasets. 44 (18%) papers used the authors' group or organizational websites, followed by Zenodo (14%) and GitHub (14%) [1] , and Google Drive(13%). There is also a long tail of less used hosting sites [2]. One unique advantage of using Zenodo, FigShare, and PhysioNet is that the dataset will be given a DOI (Digital Object Identifier), a permanent, unqiue identifier for the dataset, especially given that NeurIPS publications are not assigned any by the publisher.

Some hosting sites, like PhysioNet and Zenodo, allow users to input metadata and version information, while others, such as various cloud drives and personal websites, do not. As a result, authors may lack the incentives or ability to share metadata and maintain records of different dataset versions. This absence of metadata and version information can make it difficult for future users to understand a dataset's origin, context, and changes over time and potentially hinder their ability to reproduce results from earlier versions.

---

[1]others focusing on safety datasets have also found GitHub to be a commonly used hosting service [36]

[2]The following hosting sites were used by less than 5 dataset papers: National Center for Biotechnology Information(NIBC), OpenScienceFramework, Microsoft SharePoint, FigShare, National Institute of Health(NIH), CodaLab, Box, MLCommons, Google APIs, PyPI, OneDrive, IEEE Data Port, AIcrowd, Mendeley Data, National Cancer Institute, DropBox, OpenDataLab, and OpenICPSR.

### 3.2.1 Access: can we still access the dataset today?

The vast majority of the datasets were directly accessible via the URLs provided by the authors, without manual approvals. On Hugging Face and PhysioNet, some dataset authors required dataset users to be registered users of the sites and asked them to acknowledge their data use agreements before proceeding to the dataset.

20 papers' authors set up additional manual review procedures for those interested in accessing the dataset. Some of these are due to the sensitivity of the dataset (e.g. health records [24, 39]) or to prevent misuse (e.g. deepfakes [22] and satellite imagery [37].)

Lastly, we faced some challenges in locating the dataset URLs of some papers. In several instances, we could not find the URL in the paper or supplemental material and relied on search engines to find the dataset's hosting sites. In another case, the Zenodo link and the project link shared by the authors in the paper and supplemental material were no longer valid, but we were able to retrieve the dataset's new Zenodo link from the research team's website [3]. We also found one instance in which the download feature was disabled by its authors as of May 2024 in order to comply with policy changes [27].

### 3.2.2 Was a datasheet provided?

We saw a rather consistent usage of datasheets over the years. 48% of papers from 2021 included a datasheet. The number increased to 62% in 2022 but decreased to 53% in 2023. Because we only reviewed submitted materials for each dataset papers, datasheets shared through other means (e.g. dataset hosting platforms, arXiv.org, etc.) were not accounted in our analysis. The percentages we retrieved from our analysis may be a lower bound estimate for the adoption of datasheet among all the papers we reviewed.

### 3.3 What ethical disclosures were discussed?

96 included ethical disclosures in the paper and/or supplemental materials. We adopted Ajmani et al.'s definiteion of ethical disclosure as any reasoning in the paper or supplementary materials about ethics, potential harms, or broader impacts [1]. Below, we discuss emergent themes in the ethical disclosures we found in these dataset papers and their supplemental materials. These themes are synthesized through an iterative coding process among the two authors that annotated the papers.

*Privacy and Identification of Individuals* Privacy concerns were heavily discussed within ethical disclosure statements. Privacy was most often discussed in terms of personally identifiable information (PII). Authors often sought to protect PII by removing such information from their datasets. For example, one research team in computer vision noted they would manually inspect images for faces and license plates before publishing their dataset [30]. Authors of location-based datasets also mentioned only releasing feature-level data with appropriate credentials to prevent data leakage [41]. While discussions of PII are a necessary foundation for considering privacy, we found a lack of privacy discussion that goes beyond data content. For example, what privacy rights do we generally owe to data contributors? Does privacy demand more holistic consent procedures? These are ethical tensions within ML research that are currently lacking from the ethics discourse.

*Representativeness* We also found concerns about sampling biases of these datasets. Due to resource limitations, researchers often collected data from a single source, region, or sample (e.g. patient data from one hospital, user behavior from one region). For example, one paper noted that they used a "convenience sample of college students" resulting in potential data bias around age, race, and socioeconomic status [20]. In these cases, authors cautioned future dataset users about the potential biases and some even suggested that their datasets should only be used as an initial foundation and must be diversified before model training.

*Out-of-Context Misuse* Finally, some authors also expressed concerns about future use of these datasets and related outcomes. In sentiment analysis-focused research, authors mentioned that sentiment analysis for vaccines and drugs can only reveal the user's overall viewpoint and attitude

Table 2: Licenses used in more than 10 dataset papers.

| License | Number of dataset papers |
|---|---|
| CC BY | 61 |
| CC BY-NC-SA | 40 |
| n/a(license information unavailable) | 30 |
| MIT License | 24 |
| CC BY-NC | 23 |
| CC BY-SA | 16 |
| Apache License 2.0 | 10 |

and should not be interpreted as their willingness to take drugs or get vaccinated [44]. These more speculative ethics concerns are crucial for considering the broader impacts of research—an exercise with which NeurIPS authors have struggled in the past [32]. Future work could explore how to assist dataset authors in speculating potential misuse of their datasets.

Overall, the ratio of papers that mention ethical concerns (40%) is less than ideal. This gap suggests a need from the NeurIPS community for ethical disclosure scaffolding. In particular, unlike with models, ethical issues in datasets can be difficult to predict or foresee. Moreover, authors of dataset papers may be ill-equipped to envision the secondary, unintentional impact of dataset creation, even when prompted to do so. We urgently need frameworks that can guide authors to reflect on their dataset practices from collection to processing to aggregation to sharing.

### 3.4 What licenses were applied to datasets?

Table 2 shows the distribution of licenses used in the 238 dataset papers [3]. Creative Commons licenses are widely preferred: CC BY is the most commonly used license, with 61 papers using this license for their datasets, followed by CC BY-NC-SA (40). Notably, a non-trivial percentage of papers (72) used Creative Commons licenses that prohibit commercial use, i.e. CC BY NC, CC BY NC SA, and CC BY NC ND. In addition to Creative Commons Licenses, there are other licenses present that disallow commercial use, such as the PhysioNet Credentialed Health Data License 1.5.0.

Consistent with prior work [28], we found a notable number of dataset papers used software licenses whose suitability as dataset licenses has been questioned by some legal scholars and experts, such as the GNU GPL license, the Apache license, and the MIT license. More specifically, software licenses are designed for code and program, whereas the copyright of a dataset is more complex, especially when it contains materials whose copyrights do not belong to the dataset authors [25].

We did not find explicit license information in 36 papers, their supplemental materials, or corresponding dataset hosting sites. This lack of clarity may pose challenges for potential dataset users in determining the legal suitability of the datasets for their intended purposes.

#### 3.4.1 How easy it was to retrieve licensing information

The location of licensing information in these dataset papers was inconsistent. Out of the 238 papers, 160 (67%) included information about the specific licensing of their datasets in the appendix or supplemental materails. 54 (23%) outlined the licensing terms for their datasets in the main body of the paper. 16 papers mentioned the license information in their dataset's hosting site, e.g. under the "LICENSE" section. Some authors included licensing information multiple times at different locations. The discrepancies in where dataset licenses are disclosed will likely make it challenging for dataset users to locate accurate license information, increasing the risk of dataset misuse.

---

[3]We note that a paper may use multiple licenses to license different datasets. We also found the following licenses mentioned by less than 10 dataset papers: CC BY-NC-ND, customized license, CC0, Inheritance(same as the license of original datasets), PhysioNet 1.5.0, BSD 3-Clause, GNU General Public License, zlib, Amazon License, Community Data License Agreement, ODC-By, and NetHack GPL

### 3.4.2 Issues with copyrights

Determining who is the copyright owner of materials included in a dataset can be a challenging task, particularly when the authors collected data post hoc. For example, Ypsilantis et al. collected images from Flickr and various web sources and acknowledged the challenge of identifying individual copyright owners [43]. Due to this, the research team did not license their dataset. In another case, Wu et al. licensed their Kuaishou video dataset following the company's legal team's advice but for the videos collected from YouTube, they acknowledged that they were not the copyright holder and therefore could not license the dataset [45].

In other cases, authors licensed their datasets that contain copyrighted materials. For example, the MineDojo dataset consisted of extensive images, videos, text content from various sources including YouTube, Minecraft Wiki, and Reddit [15]. Each source has its own license or data use terms (for example, Minecraft Wiki is under CC BY-SA), and the authors applied new licenses to the aggregate datasets.

To circumvent the copyright issue, some authors opted to include URLs to original artifacts rather than the actual artifacts. For example, the LAION 5B dataset included links to various images hosted on the web and the RedCap dataset included links to Reddit's hosting service.

### 3.4.3 Efforts to retrieve content with clear copyrights

We have also observed efforts by authors to retain materials with clear copyrights when constructing their datasets. For example, Galves et al. paid special attention to licenses when crawling content from Vimeo and archive.org and only retained content with CC BY, CC BY-SA, and CC0 licenses, i.e. content allowed for commercial use [16]. The authors subsequently licensed their dataset under CC BY-SA. The authors also stated that they did not download any videos from YouTube, even the ones with permissible CC licenses, citing concerns about violations of YouTube's terms of service [16]. In a similar case, Falta et al. curated data from existing datasets, and inherited the original licenses [14].

## 4 Related Work, Discussion, and Recommendations

Through a review of 238 dataset papers, we found differing practices in sharing and managing datasets among authors. Given the diversity of the published datasets' disciplines, domains, and purposes, some differences were expected or necessary. For example, datasets that were constructed post hoc were naturally less traceable than the ones that were created ad hoc. Similarly, we expected datasets with copyrighted materials to encounter licensing uncertainties given the complex nature of these topics [26].

However, some gaps and inconsistencies warrant further reflection. Below, we discuss how stakeholders of the data supply chain, from dataset authors to publishers to institutions to funding agencies, may work collaboratively to ensure that datasets are shared and managed responsibly and ethically.

### 4.1 Needs for Standardized Data Infrastructure

Dataset authors shared datasets on a wide range of hosting sites. This aligns with previous research on National Science Foundation-funded projects [38] and evaluation datasets for LLM safety [36], which also found that dataset authors utilized a diverse array of hosting services. In particular, we saw a substantial reliance on sites that do not support metadata and version control, e.g. cloud drives and cloud hosting services. Metadata is crucial for responsible data management as it provides important context about the dataset creation process. Lack of metadata will limit future data users' ability to verify provenance, assess suitability, and potentially identify biases in a dataset. Version control is another important feature for responsible data management, as datasets expand or change, due to new data points added or old data points being corrected or deleted. Lack of version control will potentially lead to reproducibility issues in downstream models. Taken together, to ensure responsible usage of datasets and accurate models, it is important for the NeurIPS community to establish standardized

data infrastructures so that future datasets can be published with accurate, consistent metadata and version records. Such data infrastructures could: 1) have built-in templates such as dataset card or datasheet for dataset authors to input metadata in a structured format (a feature already supported to some extent by some dataset hosting platforms such as Hugging Face) and 2) display changes and contributors of different versions (a feature supported by some platforms such as dataverse.) In the meantime, publishers should also discourage dataset authors from sharing datasets without meaningful metadata or support for version tracking.

The community could benefit immediately from public and research institutes' data management services (e.g. the dataverse platform and Texas data repository). Many research fields, such as biomedical sciences and natural hazards engineering, have a long history of sharing data to advance science and foster research collaboration [7, 35]. These fields have established policies and institutional infrastructures for data sharing. For example, the National Institutes of Health in the U.S. has long invested in data sharing to support and catalyze biomedical research, including sites like PhysioNet (now funded by the National Institute of Biomedical Imaging and Bioengineering) and the National Center for Biotechnology Information–two hosting sites that have been used sparingly by the dataset papers we reviewed. Similarly, the Natural Hazards Engineering Research Infrastructure(NHERI) developed and launched its data hosting site, DesignSafe, to promote research collaborations [35]. In social sciences, data archives such as ICPSR (the Inter-university Consortium for Political and Social Research) allow researchers to deposit and share large datasets with robust documentations. These well-established data infrastructures, along with their protocols and best practices, could serve as guiding examples for publishers, research institutions, firms, and funding agencies to support similar data infrastructures for publishing ML datasets.

## 4.2 Compliance issues with the FAIR principles

Our findings on the lack of data documentation and inconsistent sharing practices also highlighted compliance issues with the FAIR principle (findable, accessible, interoperable, and reusable)–the foundational values of scientific data management [40]. Below, we unpacked the specific compliance issues and discussed recommendations for dataset authors and other stakeholders.

*Findable:* The different platforms used for dataset sharing could pose challenges to findability. Datasets hosted on prominent platforms like GitHub, Hugging Face, and Kaggle may be easier to find than those hosted on individual or group websites. Given ML researchers' concentrated usage of a selected few datasets [23], we recommend that before choosing a hosting site, dataset authors explore who their potential dataset users are and how they find suitable datasets for their work.

Additionally, dataset authors should consider the specific features offered by different hosting sites when uploading their datasets. Findability requires rich metadata and a unique, persistent identifier, which many cloud drives and hosting services currently do not support. Therefore, it may be beneficial for dataset authors to opt for hosting sites that support metadata and DOIs so their datasets can be easily retrieved and identified by search engines.

Conference organizers and publishers may also provide guidelines to help dataset authors identify and choose hosting platforms that support findability. For example, these guidelines might emphasize that links to datasets from personal or group websites may take time to be indexed by search engines, making them less findable to potential users. Additionally, given that NeurIPS papers do not have assigned DOIs, conference organizers and publishers could consider requesting dataset authors to create DOIs [4] for their datasets to ensure that these datasets can be easily found and cited. Conference organizers could even include an informational prompt beneath the DOI input field, informing dataset authors of the importance of DOIs for dataset findability and citation.

*Accessible*: Almost all datasets are accessible, either directly or after an approval process established by the authors. However, as described earlier, we encountered outdated dataset links in papers and supplemental materials. Data authors should pay special attention to the dataset links they include in their papers and the preservation of metadata and take into account that some hosting services

---

[4]If anonymity is required for peer preview, authors could use private DOIs.

may not be permanent. Consulting institutional libraries or research data management teams may be particularly helpful for dataset authors, as these information professionals have long worked toward preserving and structuring (meta)data for digital access. Dataset authors may seek specific guidance from them to identify stable hosting services, generate DOIs, and make their datasets accessible to different types of dataset users (researchers, developers, students, etc.).

Another requirement for achieving accessibility in the FAIR principles is the support of authentication when necessary [40]. After all, not all datasets should be openly accessible. In this regard, dataset authors can learn from information professionals in libraries and archives, who have managed digital access to sensitive materials for decades [9]. We found that very few hosting sites, such as PhysioNet, Hugging Face, and Zenodo, offer authentication services. Some dataset authors resorted to manual authentication processes, which can be time-consuming and labor-intensive. We recommend that authors consider the need for authentication to prevent potential misuse of their datasets and collaborate with their institutional libraries or research data management teams to identify suitable hosting sites and authentication mechanisms.

*Interoperable:* While we did not delve into the specific formats of NeurIPS datasets and how interoperable they are, we reviewed how dataset papers were built off of existing datasets. We found it challenging to trace a significant portion of these "remix" datasets due to manual sampling of original data sources or poor documentation of the curation process. Therefore, it is unlikely for such datasets to be interoperable with original data sources. We recommend dataset authors carefully evaluate whether their novel datasets need to be compatible with original data sources in their construction process and if so, share detailed preprocessing procedures to support interoperability.

*Reusable:* The wide adoption of Creative Commons licenses and open source licenses suggests that most of the datasets without copyrighted materials can be reused by other researchers. However, the different placements of license information we observed raised questions about how visible these license terms will be to potential dataset users. We recommend dataset publishers and conferences include licensing information as part of the structured metadata they collect from authors and clearly highlight this information to potential dataset authors.

Additionally, lack of provenance potentially inhibits the reusability of a dataset [40], making it all the more important to pay careful attention to dataset documentation [21], including but not limited to possible inclusion of copyrighted materials, recruitment process of annotators if applicable, and specific data collection approaches (e.g. scraping vs. API).

## 4.3   Dataset licensing

The issues of unclear copyright raise the question of whether disambiguating copyright licensing will be an effective approach for AI governance. With the deployment of large AI models, there have been extensive scholarly discourses and lawsuits on whether copyright laws could be used to govern AI models and enforce responsible reuse of creative content [26, 18]. Some researchers have also advocated for adopting licensing to restrict downstream uses of datasets and code [11]. As a result, the landscape of copyright licenses is becoming increasingly complex. However, dataset authors may be ill-equipped or reluctant to make informed licensing decisions. It is crucial for publishers to help dataset authors understand the implications of copyright licenses, as this will be an urgent next step in addressing these challenges.

Given that over half of the dataset papers fall under the post hoc category, i.e. leveraging existing data to construct a new dataset, authors would likely benefit from tools that can address licensing dependencies and recommend appropriate licenses. Such tools could be built upon existing data tracing and provenance tools such as the Data Provenance Explorer [28] and the What's In My Big Data? platform [13] and provide parental licenses of original datasets so authors could make informed decisions about who are the copyright holder and whether their new datasets need to inherit any parental licenses.

### 4.4 Limitations and Future Work

Our review focused on the NeurIPS Datasets and Benchmarks track and did not include any dataset papers that have been published in the main conference track. Nonetheless, given the breadth of our review, we expect our sample to reflect the lack of consistency in dataset management practices in the NeurIPS community. Additionally, given the track's detailed submission requirements, it is likely that dataset documentation, licensing, and ethical disclosures are reported in greater details than they are at other publication venues. Future work may expand our review by including ML datasets from other venues to gain a more holistic understanding of dataset management practices in the field.

Our review also is limited to the four aspects of data management practices (provenance, distribution, ethical disclosures, and licensing) and does not consider secondary impact of these datasets, e.g. potential violation of original creators' copyrights and moral rights and distribution of harmful content–a fruitful area for future work [34].

Our findings and recommendations aim to inform the development and adoption of standardized data infrastructures and data management practices and do not serve as legal advice. Moreover, as mentioned earlier, we caution against blanket data standards applied to different types of datasets. For example, For instance, datasets containing sensitive materials or those that could be misused may require more rigorous stewardship and stricter standards compared to datasets that consist solely of public content.

We shared our detailed annotations on which future work could build to automate part of our review. Interested researchers should pay particular attention to several significant challenges for automation that we have identified: unclear terms and descriptions regarding data collection approaches, inconsistent naming and locations of licenses, different methods of sharing dataset URLs, and varying levels of ethical disclosures. Some of these challenges related to retaining licensing information and URLs could be addressed promptly if NeurIPS guidelines provided a uniform, structured format for authors to follow.

## 5 Conclusion

We reviewed 238 dataset papers published at the NeurIPS Datasets and Benchmarks track between 2021 and 2023 and examined the provenance, distribution, ethical disclosures, and licensing of the output datasets. We found inconsistent practices across all aspects, discussed the importance of standardized data infrastructures and compliance with the FAIR principles, and provided corresponding recommendations.

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

Figure 1: Year to Year Comparison - Hosting Domains

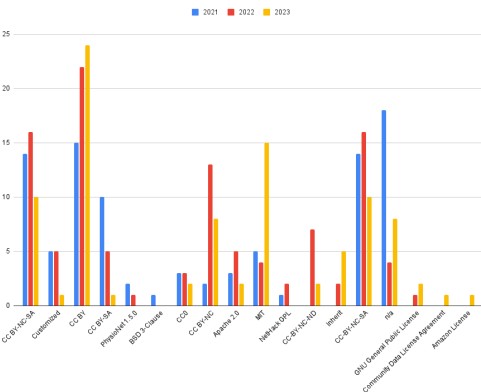

Figure 2: Year to Year Comparison - License

5. If you used crowdsourcing or conducted research with human subjects...

   (a) Did you include the full text of instructions given to participants and screenshots, if applicable? [N/A]

   (b) Did you describe any potential participant risks, with links to Institutional Review Board (IRB) approvals, if applicable? [N/A]

   (c) Did you include the estimated hourly wage paid to participants and the total amount spent on participant compensation? [N/A]

## Appendix

Figure 1 visualizes the major hosting sites across the year 2021 to 2023. We only included major hosting sites used by more than five dataset papers. We saw that personal or lab websites were a common choice in 2021 but became less common in 2023. Conversely, we saw an increasing number of datasets hosted on Google Drive, Hugging Face, and AWS in 2023. Note that given the limited time span of the datasets in our review, the current year-to-year comparison was only descriptive and should not be interpreted as statistically meaningful.

Figure 2 visualizes the licenses applied to the datasets from 2021 to 2023. The CC-BY license appeared to be the most common license across three years. Again, given that we only reviewed datasets from three cycles, differences in license usage across years should be interpreted with caution.

