# OpenReview forum: "A Systematic Review of NeurIPS Dataset Management Practices"
_NeurIPS.cc/2024/Datasets_and_Benchmarks_Track — NeurIPS 2024 Track Datasets and Benchmarks Poster_

### Official Review · Reviewer_5eR4 · 2024-06-29
**Interesting survey of data provenance**

**Rating:** 7
**Confidence:** 4
**Clarity:** The paper is clear and well written.

**Review:**

The quality of the paper is commendable, presenting a thorough and systematic review of current practices in dataset management. The clarity is evident in the well-structured analysis and comprehensive discussion of findings. The originality lies in its large-scale audit of NeurIPS D&B papers, a novel approach to understanding dataset management practices at a major machine learning conference. The significance is substantial, as it addresses critical issues in the ethical and legal management of datasets, which are foundational to ML research.

Pros:
   -Extensive dataset: The review covers a large number of papers (239), providing a broad overview of practices.
   -Comprehensive analysis: The focus on provenance, distribution, ethical disclosure, and licensing offers a detailed examination of key aspects of dataset management.
   -Recommendations: The paper provides actionable recommendations for improving dataset management practices, which could significantly impact future research.
Cons:
   -Limited scope: The review is limited to NeurIPS dataset papers, which may not fully represent practices in other conferences or disciplines.
   -Variability in documentation: Inconsistencies in how datasets are documented and shared might limit the generalizability of the findings.

**Strengths:**

The primary strengths of the submission include its significant contribution to understanding and improving dataset management practices. The systematic approach and comprehensive coverage of dataset papers make it a valuable resource for the broader research community. The quality of research is high, supported by a rigorous methodology and detailed analysis. The paper's emphasis on ethical and social implications is particularly relevant, addressing the critical need for responsible data provenance and management.

**Additional Feedback:**

N/A

**Correctness:**

The claims made in the submission are well-supported by the data and analysis presented. The methodology is sound, with clear criteria for selecting and categorizing the dataset papers. The evaluation methods and experiment design are appropriate and I'm comfortable with the reliability of the findings.

**Documentation:**

How do you plan to maintain the dataset after submission? The airtable is very useful and informative, serving as a helpful tool for researchers. Its availability and intended future use would be important to the contributions of this paper.

**Ethics:**

The authors have addressed key ethical concerns and provided valuable guidance for the research community. It provides recommendations for addressing these issues, such as improving ethical disclosures and considering the broader impacts of datasets. These insights are crucial for ensuring responsible and fair use of data in machine learning research.

**Limitations:**

The authors have adequately addressed the limitations and potential negative societal impacts of their work. The acknowledgement of the limited scope of NeurIPS D&B track and variability between papers is important.

However, they could further emphasize the importance of considering the secondary and unintentional impacts of datasets, as well as the need for ongoing updates to dataset documentation and management practices. Constructive suggestions for improvement include developing frameworks to guide authors in reflecting on their dataset practices and incorporating ethical considerations throughout the dataset lifecycle.

**Opportunities For Improvement:**

While the paper provides a thorough analysis, it could benefit from expanding its scope to include dataset management practices in other conferences and disciplines. This would enhance the generalizability of the findings.

Additionally, providing more detailed case studies or examples of best practices could help illustrate the recommendations and support their adoption by researchers. For example, I would be interested if you have specific observations on papers that do have strong discussion of ethics, FAIR principles, and the differences in licences. For example, are those with customized licences designed because of a better consideration of ethical issues? Assessments and links between the various observations would strengthen your analysis and discussion.

**Relation To Prior Work:**

Overall, the authors deal well with related work and where they draw their methodologies, definitions and frameworks for the survey.

I would be curious if similar surveys on data provenance have been conducted over other bodies of work or repositories? If so, comparison of findings would be helpful and, if not, it would help to highlight the novelty of this work.

**Summary And Contributions:**

The paper, titled "Norms for Managing Datasets: A Systematic Review of NeurIPS Datasets," provides an extensive review of dataset management practices within the NeurIPS Datasets and Benchmarks track. The authors examine 239 dataset papers published between 2021 and 2023, focusing on four key aspects: provenance, distribution, ethical disclosure, and licensing. They identify significant inconsistencies in these areas, highlighting the need for standardized data infrastructures and improved guidelines for ethical and legal dataset management. This systematic review aims to support the responsible and ethical sharing, management, and use of datasets within the ML community.

---

### Official Review · Reviewer_rjqg · 2024-07-22
**A systematic review begging for automation.**

**Rating:** 6
**Confidence:** 5

**Review:**

### Summary
This paper makes the important effort to collect and analyze data on the dataset-management practices adopted by the authors of dataset papers on the NeurIPS Datasets and Benchmarks track. The data collected enable the track to perform crucial self-reflection, and the results of their analysis can inform improvements of the track going forward. However, the data-collection process adopted was manual (although it is often "grunt work" that can be readily automated), the sample is limited, and the data analysis is superficial. If these shortcomings are remedied (see my suggestions in other parts of this review), I will support publication of this manuscript on the NeurIPS Datasets and Benchmarks track.

### Strengths
- (S1): Dataset-management practices are an important area of self-reflection for the track

### Weaknesses
- (W1): Limited sample size (see "Correctness" below)
- (W2): Manual approach yields one-off results that are hard to replicate, generalize, and scale; no discussion of potential for or barriers to automation (see "Limitations" and "Correctness" below)
- (W3): Shallow discussion of results in terms of what they mean for the track – you devote considerable space to section 4 but end up mostly reiterating general shortcomings and recommendations already identified elsewhere. I would like to see a closer integration of this section with your own results, highlighting and elaborating on your most problematic findings. For example:
  - Dataset hosting: This is as bad as I expected, and anything that does not support DOIs and versioning presents a huge issue for reproducibility. This is only aggravated by dataset loaders in popular ML libraries following similarly bad practices (such as downloading from unstable URLs), which you might want to mention. It is also very easy to automatically check for in the review and publication process.
  - The role of the review process: To which extent do your findings indicate failures in the track's review process? What checks could be automated in the process, what checks could be performed by reviewers as part of the review process, in which areas would reviewers need better education to enforce _best_ data-management practices?
  - Post-hoc datasets: What does it say about the track if most datasets published here are post-hoc re-hashes of previously existing datasets? (My impression is that the track is often used by authors to get "cheap NeurIPS publications" – many people even "forget" the track suffix when referencing their publications. The prevalence of post-hoc datasets, which are the lowest effort of all options, seems to corroborate this impression...?)
  - "Continuous integration" of (a semi-automated version of) the systematic review into the track's review and publication process: Can you develop a vision here that would eliminate the need to do the systematic review manually and update it only periodically and also make the results more visible for anyone interacting with the track's published papers – e.g., could the track have a dashboard like your AirTable as a continuously-updated overview, plus the checklist next to each published abstract?
- (W4): Your current title is "Norms for Managing Datasets: A Systematic Review of NeurIPS Datasets". I know that you use the term "norm" to mean "something that is usual" (understood empirically), but it is unfortunately close to "normative", which is commonly used to contrast with "empirical" and to embody _desired_ practices (which are clearly not the norm in our community yet). Hence, I would recommend eliminating "norms" from the first part of the title (e.g., by substituting it for "Practices" or rewriting the title as "Dataset Management in the Wild: ...").

**Strengths:**

See above.

**Additional Feedback:**

- Could you check if the temporal analysis (dataset papers split by years) reveals any interesting trends?
- You might want to highlight the potential positive impact of the transparency enabled by your work (and its generalization) on best-practices adoption rates.
- Your references are currently included twice.
- Check the first author in reference [47] (which currently starts ", ...").

**Clarity:**

The paper is easy to follow, but it discusses statistics mostly in natural language where graphics would be more effective (for example, many of your paragraphs would be more telling as bar charts).
I encourage the authors to consider supplementing their text with figures and then shortening the natural-language descriptions as appropriate.
You might also want to a PRISMA-style diagram going from "all NeurIPS D&B papers published on the track before 2024" to your actual sample, as is common in systematic reviews.
Finally, there are some typos and missing spaces as well as hyphenation and word-choice issues, so you might want to make another editorial pass over the manuscript.

**Correctness:**

### Sample Composition
You write:
> Because 2023 saw substantial growth in number of papers, we randomly sampled 80 out of a total of 193 published.

Why exactly did you sample here? Sure, handling each dataset paper takes time, but given the numbers we are looking at here, this appears pretty doable (I know it is tedious, but if the constraint here was time, it all the more demonstrates the need for automation).
If your concern is a temporally balanced sample, I would rather annotate all papers and then do statistics for each year separately.
This would also allow you to analyze if practices changed over the years (an interesting and important question that you could already tackle, at least partially, with your existing annotations).
Could you extend your analysis to contain all published datasets papers on the track?

### Generalization
In the context of datasheet adoption, you mention that your percentages could be a "lower bound" on the adoption in the NeurIPS community.
I suspect that either you mean "upper bound" here (if the reference to the general NeurIPS community is correct) or you mean "lower bound on the adoption in our sample".
In "Limitations", you write that "given the number of papers in our scope, we expect our sample to reflect the general trend of dataset management practices [in the broader NeurIPS community]".
Again, "upper bound" is more realistic (as in: practices in the community can be expected to be no better than on the track).

**Documentation:**

Please deposit the data currently in the AirTable file as a research-data artifact in a repository assigning DOIs (e.g., Zenodo), and include the DOI to this artifact in the paper.
Furthermore, the description of your annotation procedure is very terse (ll. 63–67), and the process is hardly reproducible given your current description.
Could you elaborate, and also explain if/why you deviated from established annotation best practices?

**Ethics:**

No significant concerns.

**Limitations:**

The discussion of limitations is rather shallow, and one of the biggest limitations is not addressed at all:
The lack of scalability inherent in the manual-annotation approach, and the corresponding need for automation.
I encourage the authors to be explicit about this limitation, and to develop a vision for (semi-)automating their systematic review process to enable continuous, low-maintenance updates and full integration with the review process of the track.

**Opportunities For Improvement:**

See above.

**Relation To Prior Work:**

The discussion of related work is very limited (the corresponding heading reads "Related Work, Discussion, and Recommendations", but there is hardly any discussion of related work in that section). In particular, the paper would benefit from contextualization with respect to the following discourses:
- empirical and taxonomy-building work on (ir)reproducibility in machine learning
- principles of open science and FAIR data (esp. to contrast with your empirical findings)
- empirical work on research-data-sharing practices in other fields (to compare with your empirical findings)

**Summary And Contributions:**

The paper presents a systematic review of datasets published on the NeurIPS Datasets and Benchmarks track, examining their dataset-management practices with respect to provenance, distribution, ethical disclosure, and licensing.

---

### Official Review · Reviewer_jDR7 · 2024-07-22
**Strong paper with room to improve**

**Rating:** 7
**Confidence:** 3
**Clarity:** The paper is clearly written.

**Review:**

Overall this paper provides clear, relevant and timely contributions to the D&B track.

**Strengths:**

The evaluation is a clear strength of the paper. These findings are of great interest to the D&B community and highlight glaring problems within the track despite the increased understanding of the importance of datasets and their management.

The paper benefits from having a clear scope, by addressing the four elements mentioned above.

The paper also provides clear and actionable recommendations as to how the D&B track can begin to address these challenges.

**Additional Feedback:**

Some additional points that I had were:
- The authors mention version control as a key issue within dataset management. Of the datasets investigated, how many had changed since they were published? How were those changes communicated?
- I appreciated the suggestion to standardize dataset infrastructures, e.g., rich metadata and unique identifiers (298) that many dataset platforms do not support. Do the authors suggest any specific platforms or infrastructures that allow for these things?\
- Please note the typo on line 355 "NeruIPS"
- The paper mentions that the evaluation schema was developed by going through 30 papers. Were there elements that were considered but not included in this analysis? For instance, looking at the annotations it seems that reproducibility was a concern at some point for the authors. However, this category is unclear. Some forms of data collection don't have any reproducibility, while some that do have little detail. Is there a difference between 'possible' and 'yes'?

**Correctness:**

The claims made are correct, so far as I can tell. The paper does not provide a dataset.

**Documentation:**

Not a dataset submission. Nevertheless, the methodology is largely well explained with annotations publicly available.

**Limitations:**

The limitations section is very brief and has great opportunity for further reflection. Some considerations are above, but I encourage the authors to think beyond these.

**Opportunities For Improvement:**

While this is a strong paper there are some areas for improvement:
- A limitation of the paper is the limitations section itself, which has much greater opportunity for expansion.
- How might the sampling of the datasets affect the results of this review?
- What other key issues regarding dataset management might be missing from this review? Why were these four key elements (and the specific sub-elements) chosen? What was not chosen to investigate?
- The limitations briefly mention cautioning against "blanket data standards" (352) but this is the place to unpack this in greater depth.  What are some potential drawbacks of standardization of documentation practices?
- The paper does not mention any proposed methods for standardizing ethical disclosures, despite the observed variability. A reflection on how this could be standardized, or why it should not be standardized would be beneficial. The paper calls for "ethical disclosure scaffolding" but more detail regarding what this might look like would be beneficial.

**Relation To Prior Work:**

This paper cites existing work well and connects with broader discussions of dataset management.

**Summary And Contributions:**

This paper reviews 239 dataset papers published in the Neurips D&B track between 2021 and 2023, focusing on four key elements of data management practices: provenance, distribution, ethical disclosures, and licensing. The authors identify some key issues regarding dataset management within the D&B track such as unclear data provenance, varied and inconsistent distribution platforms, inconsistent ethical disclosures and often incomplete licensing information. This paper therefore illustrates the need for further standardization of dataset management practices and infrastructures to safeguard processes of transparency.

---

### Official Review · Reviewer_55ZR · 2024-07-25
**In-depth analysis but little insight into what to take away from it**

**Rating:** 6
**Confidence:** 4
**Correctness:** The claims seem to be supported by ev…

**Review:**

See below.

**Strengths:**

- The paper provides a comprehensive review of a large sample of recent NeurIPS Dataset & Benchmark track papers, providing valuable insights into current practices, especially within the community that they are submitting the paper to
- The paper designs a structured analysis including multiple important dimensions of data management and executes that analysis
- The authors highlight critical issues around dataset provenance, ethical considerations, and licensing that have significant implications for responsible AI development and deployment
- The authors connect their findings back to established principles like FAIR and relating their work back to existing best practices

**Additional Feedback:**

N/A

**Clarity:**

The paper is very well written, except a few smaller typos (e.g., "definitieon" and some missing prepositions)

**Documentation:**

Analysis data is available online. No additional documentation necessary.

**Ethics:**

No.

**Limitations:**

- The authors note that their review focused only on the NeurIPS Datasets and Benchmarks track, potentially missing datasets published elsewhere. I missed an additional discussion here on the bias that this introduces. For example, this track specifically recommends the use of data cards or similar documentation, which may bias results by having a larger number of well-documented datasets than it would occur otherwise; also, for the analysis of use of data cards over time, it would be good to include since when NeurIPS recommended these data cards. I also missed a statement saying that because of such biases, the results do not necessarily generalize to the broader ML community beyond NeurIPS.
- The authors acknowledge that some differences in practices are expected or necessary given the diversity of datasets and that this is the reason that they do not provide standardized recommendations but to me, it is unclear what to make of these results/how to actually improve this situation in the community.

**Opportunities For Improvement:**

- (Biggest concern) The authors could expand on potential solutions or frameworks to address identified issues. In the current analysis, the recommendations or change suggestions are very brief and the reader is left wondering what to do about the identified issues. E.g., the recommendation "Consulting institutional libraries or research data management teams may be useful for dataset authors" is not directly actionable. What questions should they ask? On what exactly should these organizations be consulted? Is there an example you can provide? I understand the caveat that the authors mentioned on the diversity of datasets and that this makes general standards tricky, but I find the current recommendations too high level to be implemented by concerned parties and to really drive change.
- The paper could benefit from more quantitative analysis of trends over the 3 year period examined (similar to brief analysis of the adoption of data cards over this time)
- The authors could further discuss implications of findings for reproducibility and accountability in ML research
- (Likely out of scope for this paper, no need to address this in detail) The authors could explore incentive structures or policy changes in more detail that could drive adoption of better practices

**Relation To Prior Work:**

From Section 4, it's somewhat unclear to me to what extent similar analyses were done before. They mention that there was "prior work focusing on data management plans" but it's unclear what that covered. It would be great if the authors could add other previous similar analyses – if they exist – or specifically state that they are (to the best of their knowledge) the first that performed this kind of work.

**Summary And Contributions:**

This paper presents a systematic review of 239 dataset papers published in the NeurIPS Datasets and Benchmarks track from 2021-2023. The authors examine four key aspects of data management practices: provenance, distribution, ethical disclosure, and licensing. They find major inconsistencies across all four aspects. Key contributions include identifying gaps in dataset provenance tracing, cataloging the variety of hosting platforms used, analyzing the extent of ethical considerations discussed, and examining licensing practices.

---

> ### Comment · Reviewer_55ZR · 2024-08-31
> **Response to author feedback**
>
> Thank you for providing additional details and clarifications to my concerns and improvement suggestions. Would you be able to upload the improved recommendations as pdf (or post them as response to this comment) so that I can take a look at them and see if they were sufficient for a score increase? Thanks!

---

> > ### Author Response · Authors · 2024-09-04
> > **recommendation section**
> >
> > Thank you for your interest in our revised manuscript! We are still not sure if we are allowed to share the revised manuscript as a PDF (see our comment above on NeurIPS's guidelines on rebuttal), but we believe we can share specific improvements here.
> >
> > Due to the character limit here, we only included paragraphs that have new or extended recommendations. We hope this helps. Thank you again for your time and engagement!
> >
> > ------
> > Needs for Standardized Data Infrastructure
> > …Taken together, to ensure responsible usage of datasets and accurate models, it is important for the NeurIPS community to establish standardized data infrastructures so that future datasets can be published with accurate, consistent metadata and version records. <em>Such data infrastructures should: 1) have built-in templates such as dataset card or datasheet for dataset authors to input metadata in a structured format (a feature already supported to some extent by some dataset hosting platforms such as Hugging Face) and 2) display changes and contributors of different versions (a feature supported by some platforms such as dataverse.) In the meantime, publishers should also discourage dataset authors from sharing datasets without meaningful metadata and support for version tracking. <em>
> > …
> >
> > Compliance issues with the FAIR principles
> > Findable: …
> > Findability requires rich metadata and a unique, persistent identifier, which many cloud drives and hosting services currently do not support. <em> Therefore, it may be beneficial for dataset authors to opt for hosting sites that support metadata and DOIs so their datasets can be easily retrieved and identified by search engines. </em>
> >
> > Accessible: … Consulting institutional libraries or research data management teams may be particularly helpful for dataset authors, as these information professionals have long worked toward preserving and structuring (meta)data for digital access. <em>Dataset authors may seek specific guidance from them to identify stable hosting services, generate DOIs, and make their datasets accessible to different types of dataset users (researchers, developers, students, etc.). </em>
> >
> > Another requirement for achieving accessibility in the FAIR principles is the support of authentication when necessary [24]. After all, not all datasets should be openly accessible. <em>This is another area where dataset authors may learn from information professionals from libraries and archives who have been stewarding digital access to sensitive materials for decades [8]. </em> We found that very few hosting sites used by dataset papers offer this service, i.e. PhysioNet, Hugging Face, and Zenodo, and some dataset authors required a manual authentication process. We recommend authors consider whether authentication would need to be required in the future in case of dataset misuse <em>and work with their institutional libraries to identify appropriate hosting sites and authentication mechanisms. </em>
> >
> > Interoperable: … We recommend dataset authors carefully evaluate whether their novel datasets need to be compatible with original data sources in their construction process <em> and if so, share detailed preprocessing procedures to support interoperability.  </em>
> >
> > Reusable: Lack of provenance also inhibits the reusability of a dataset [12], making it all the more important to pay carefully attention to dataset documentation [35]. <em> We recommend dataset authors include detailed information about provenance in their documentation, including but not limited to possible inclusion of copyrighted materials, recruitment process of annotators if applicable, and specific data collection approaches (e.g. scraping vs. API). </em>

---

### Author Rebuttal · Authors · 2024-08-17

We thank the reviewers for their constructive feedback and comments. We are glad that the reviewers find our results “are of great interest to the D&B community” and believe that “The systematic approach and comprehensive coverage of dataset papers make it a valuable resource for the broader research community.” After reading the reviews, we were convinced that it was important to unpack our recommendations, expand our limitation section, and discuss opportunities for future work, as well as adding clarifications.

- Elaborate on recommendations: All reviewers asked for details about our recommendations for dataset management practices based on our findings. Reviewer jDR7 raised specific questions about ethics scaffolding. We reworked our recommendation section to include related taxonomies (e.g. Andrews et al. 2023) and examples of best practices from our review. Similarly, we unpacked other recommendations such as “consulting institutional libraries or research data management teams” and included specific considerations and measures dataset authors may take into account (e.g. what data hosting services allow authors to track the download and usage of their dataset).

- Expand limitations: We agreed with Reviewer 5eR4 and Reviewer 55ZR that our choice of venue was a limitation. We also included additional limitations such as our focus on only four aspects of dataset management and the limited time span of these datasets (see more below). Additionally, we agree with Review 5eR4 that our current focus did not include ethical considerations of the potential secondary impact of datasets (e.g. copyright violation and sharing harmful content) as a limitation of our work.

- Potential for Automation: Reviewer rjqg suggested an automated process to expand the scope of our work. We agree that for the quantifiable measures we examined, automation will be a very plausible next step; in fact, we were motivated to do this “grunt work” partially because of our own curiosity about to what extent dataset metadata is structured and machine-readable. Our investigation revealed several significant challenges for automation in the current landscape – unclear descriptions about data collection approaches, inconsistent naming and locations of licenses, differing ways of sharing dataset URLs, and varying degrees of ethical disclosures. Notably, these findings are qualitative in nature and would not have surfaced with an automated approach. Moreover, taking an automated approach necessitates a buy-in from NeurIPS organizers. Accordingly, we expanded our recommendations to include specific changes to the current publishing system so that future work can automate our investigation.

- Contribution and Scope: Relatedly, Reviewer rjqg asked why we randomly sampled papers from the 2023 batch. We clarified that our primary, intended contribution is to unpack “in what ways” dataset management practices differ, rather than “how many”, given that the D&B track is new and rapidly growing. While we included quantifiable measures in our writing, the numbers are to provide context and augment our qualitative findings. With this primary contribution in mind, we began with sampling 20 papers of the 2023 batch. We iteratively coded papers until we reached qualitative saturation and temporal balance, resulting in 80 papers from 2023. We made these clarifications accordingly in Introduction and Methods.

- Year-to-Year Comparison: Reviewers 55ZR and rjqg suggested to compare our findings by year. We produced an Appendix including year-to-year comparison of access, hosting platforms, and licenses used. We cautioned that because we only had datasets from three cycles, the current year-to-year comparison was only descriptive and should not be interpreted as statistically meaningful.

Other clarifications:
- Why the four aspects: Reviewer jDR7 asked why we specifically chose these four aspects. We revised Introduction to clarify that our choices are based on the emergent need for dataset stewardship (Borgman et al. 2024), coupled by the track’s exponential growth in the past year. Put another way, these four areas are where stewardship from publishers and stakeholders can make an immediate impact on dataset management practices.
- Lower bound estimates: Review rjqg asked how our estimates were lower bounds. We clarified that for datasheet adoption rate, our findings were the lower bounds of the reviewed papers from the D&B track (rather than “the NeurIPS community” as we had originally written). Some of the datasets might have datasheets that were not part of the submission, so they were not accounted for by us. However, it is possible that our findings are the upper bounds of ML datasets in general, because of NeurIPS’s requirement of dataset documentation and ethics disclosures.
- Related work: We expanded our related work section to better contextualize our work in relation to prior work (e.g. Sharma et al. 2023) and include more examples of how other disciplines adopted the FAIR principles.
- Title: We thank Reviewer rjqg for their comment on our title. We agreed that our title may imply normative practices and updated our title to *Norms for managing datasets, or a lack thereof*, to more accurately reflect our findings, though we are open to suggestions.
- AirTable: We are currently working with our university library to deposit our annotations into an archival repository as the AirTable link might expire in the future. We would include the permanent URL in our camera ready, should this work be accepted.
- We would also conduct additional proofreading to correct typos and references errors before submitting our camera ready.

Again, we’d like to thank the reviewers for their positive assessment and helpful feedback and comments that strengthened our paper. We believe and hope that these changes have addressed core feedback—and let us know otherwise!

---

> ### Comment · Reviewer_rjqg · 2024-08-23
> **Response to Rebuttal Comments**
>
> Thank you for responding to the questions and concerns raised by the reviewers.
>
> Your rebuttal mentioned a couple of changes you made to the manuscript, but I could not locate the amended version of the manuscript.
> Could you kindly provide (a link to) the current version?
>
> Regarding the title:
> I would prefer something that doesn't use the term "norm" because you are solely documenting behavior (is) and not expectations or anything else that could be considered normative (ought), so speaking of "norms" is misleading.
> One potential alternative: "How are ML datasets managed in practice? A systematic review of NeurIPS datasets".
> Or really anything that avoids normative connotations and emphasizes "practice" or "empirical" in the title (the subtitle is okay).

---

> > ### Author Response · Authors · 2024-08-27
> >
> > Thank you for the suggestion! We will remove "norm" from the title accordingly.
> >
> > Re: the current version of our manuscript, we are currently working with our ACs to see if it is possible to share it at the discussion stage. For the rebuttal, we followed the instructions provided by the NeurIPS paper track that asked us not to include a new version of the manuscript or any links: https://neurips.cc/Conferences/2024/PaperInformation/NeurIPS-FAQ
> >
> > But if there is anything particular we could clarify in our rebuttal, happy to provide more details!

---

> > > ### Comment · Reviewer_rjqg · 2024-08-30
> > > **Response to Additional Author Comments**
> > >
> > > Thank you for responding to my additional comments.
> > >
> > > Regarding the manuscript, I did not mean to ask you anything that violates the rules of the track (I encountered this restriction in other review processes in the past and always forget which fora it applies to), so I understand if you would rather not share the updates.
> > >
> > > (I am not sure what the rationale behind the rule is, but since I have seen many review processes in which authors made promises on which they later did not follow through, I normally prefer seeing the implemented changes, which also enables feedback on the implementation.)

---

> > > > ### Author Response · Authors · 2024-09-04
> > > > **Attaching appendix**
> > > >
> > > > Thank you for your response! Based on the instructions provided by NeurIPS, we believe we can share new figures and specific improvements here.
> > > >
> > > > We have uploaded our appendix with new figures on year-to-year comparison to this link: https://drive.google.com/file/d/1beLkiQnKwS9pkimUtq9Sco_joB0EPUo8/view?usp=sharing
> > > >
> > > > If there are any specific sections we could share, we are more than happy to paste them here as a comment. A quick note that reviewer 55ZR asked about our recommendation section, and we have pasted it below. Please feel free to let us know if there is any other specific section you would like to review.

---

### Decision · Program_Chairs · 2024-09-26

**Decision:**

Accept (Poster)

**Comment:**

This paper conducts a systematic review of 239 datasets published in the NeurIPS D&B track between 2021 and 2023, with a particular focus on dataset management practices. The findings include common issues in tracing dataset provenance as well as inconsistencies in dataset hosting and version controllability.

Reviewer opinions on this paper all lean positive, with final overall ratings of 7, 7, 6, and 6. Review quality is fairly high, with each review providing extensive feedback on different aspects of the paper. The authors provided a detailed unified rebuttal to all reviewers. Only one reviewer engaged with this rebuttal, maintaining their score.

Overall, I believe reviewer assessments to be well-founded. Most reviewers commend the clarity of the paper’s structure and writing, as well as the scope of the survey. There is a trend towards better (unified) documentation of all parts of the dataset development process, and this paper makes a valuable contribution towards this trend by focusing on dataset management practices.

On the more negative side, I agree with reviewer 55ZR that it is difficult to take concrete actions based on the findings of the paper. The recommendations made towards the end of the paper are vague. They could perhaps be made more concrete by, for example, proposing expansions for an existing dataset reporting framework, like [Data Statements]( https://aclanthology.org/Q18-1041/). For ease of reading, I would also consider making Related Work a separate section from Discussion and having Recommendations as an explicit (sub)section.

Overall, based on reviewer comments and my own review of the paper, I recommend acceptance for poster presentation.

Minor notes:
- I would have been curious to see at least a small stat on the modalities of the datasets, since this may also affect how they are / how they can be shared.
- The [SafetyPrompts review]( https://arxiv.org/abs/2404.05399) of open datasets for LLM safety may be an interesting related works reference, since it also reviews dataset licensing and hosting.

On the title:
- The most descriptive paper title would probably be “A Systematic Review of Current Norms and Practices for Managing NeurIPS Datasets”, but that does not sound very snappy or exciting.
- I agree with reviewer rjqg that the paper focuses on “Management Practices” more than “Norms”.
-  I would also consider making explicit that there are practices for managing *ML* datasets.

Formatting:
- Stylistically, using “such as” rather than “e.g.” in regular sentences is more elegant (see Intro)
- There are typos in the caption for Table 2.